# Current Trends in the Reconstruction and Rehabilitation of Jaw following Ablative Surgery

**DOI:** 10.3390/cancers14143308

**Published:** 2022-07-07

**Authors:** Jane J. Pu, Samer G. Hakim, James C. Melville, Yu-Xiong Su

**Affiliations:** 1Division of Oral and Maxillofacial Surgery, Faculty of Dentistry, The University of Hong Kong, Hong Kong; u3503024@connect.hku.hk; 2Department Oral and Maxillofacial Surgery, University Hospital of Lübeck, Ratzeburger Allee 160, 23538 Lübeck, Germany; samer.hakim@uni-luebeck.de; 3Department of Oral and Maxillofacial Surgery, University of Texas Health Science Center at Houston, Houston, TX 77030, USA; james.c.melville@uth.tmc.edu

**Keywords:** jaw reconstruction, microvascular reconstruction, free flaps, computer-assisted surgery, patient-specific implants, virtual surgical planning, 3D printing

## Abstract

**Simple Summary:**

The Maxilla and mandible provide skeletal support for of the middle and lower third of our faces, allowing for the normal functioning of breathing, chewing, swallowing, and speech. The ablative surgery of jaws in the past often led to serious disfigurement and disruption in form and function. However, with recent strides made in computer-assisted surgery and patient-specific implants, the individual functional reconstruction of the jaw is evolving rapidly and the prompt rehabilitation of both the masticatory function and aesthetics after jaw resection has been made possible. In the present review, the recent advancements in jaw reconstruction technology and future perspectives will be discussed.

**Abstract:**

The reconstruction and rehabilitation of jaws following ablative surgery have been transformed in recent years by the development of computer-assisted surgery and virtual surgical planning. In this narrative literature review, we aim to discuss the current state-of-the-art jaw reconstruction, and to preview the potential future developments. The application of patient-specific implants and the “jaw-in-a-day technique” have made the fast restoration of jaws’ function and aesthetics possible. The improved efficiency of primary reconstructive surgery allows for the rehabilitation of neurosensory function following ablative surgery. Currently, a great deal of research has been conducted on augmented/mixed reality, artificial intelligence, virtual surgical planning for soft tissue reconstruction, and the rehabilitation of the stomatognathic system. This will lead to an even more exciting future for the functional reconstruction and rehabilitation of the jaw following ablative surgery.

## 1. Introduction

The maxilla and mandible play a crucial role in maintaining the facial contour and the upper airways and ensure masticatory function and the lip seal for swallowing and speech articulation. Ablative surgery of the jaw, for various reasons, often leads to a significant compromise of the jaw’s function and aesthetics, requiring reconstructions that are technique-sensitive and often time-consuming.

In recent years, the development of computer-assisted surgery and patient-specific implants have revolutionized jaw reconstructive surgery. With their high efficiency, accuracy, and predictability, these techniques have pushed the boundaries of functional jaw reconstruction further [1]. The development of the jaw-in-a-day technique has demonstrated a great capability for restoring the aesthetics and masticatory function at the same stage of ablative surgery. The improved efficiency of surgery also brought new opportunities for restoring the neurosensory function of the lower face.

Recently, a great deal of research has been undertaken on augmented/mixed reality surgery, artificial intelligence, virtual surgical planning for soft tissue reconstruction, and muscular and stomatognathic system rehabilitation.

In this narrative literature review, we aim to discuss the current state-of-the-art of the reconstruction and rehabilitation of the jaw and preview the potential future developments. These ground-breaking publications were summarized with the supplementation of papers yielded from keyword searches of each field in Google Scholar, PubMed, and ScienceDirect.

## 2. Current Development

### 2.1. Computer-Assisted Surgery (CAS) and Virtual Surgical Planning

Computer-assisted surgery has revolutionized jaw reconstructive surgeries. Through virtual surgical planning, more predictable and accurate jaw reconstructions can be achieved. In addition, a significant amount of operative time can be saved, thus enabling complex functional jaw reconstruction.

#### 2.1.1. Procedures of Computer-Assisted Surgery

Computer-assisted surgery normally starts from the clinical history-taking and physical examination. The history of previous surgeries and radiation therapy to the head and neck is crucial for the selection of suitable recipient vessels. A clinical examination is performed to determine the nature and extent of the lesion and formulate a tentative surgical plan. An intraoral scan is obtained to register the occlusion and the intraoral extension of the lesion. A CT scan of the head & neck and a CT angiogram of the donor site are crucial for assessing the vessel’s condition and the suitability of the free flap donor site.

Three-dimensional models of the donor and recipient sites are built based on the CT and intraoral scans. Virtual resections and reconstructions may be performed for a better understanding of the intraoperative condition. Based on the virtual planning, surgical positioning, cutting guides, and/or patient-specific implants are designed. (Figure 1).

A post-operative evaluation that compares the virtual surgical plan and the final surgical outcome is crucial for the assessment of the accuracy of the computer-assisted surgery and potential future improvements [1].

#### 2.1.2. Advantages of Computer-Assisted Surgery

Since the surgical procedure is well planned before the operation, the efficiency of surgery significantly improves with CAS. The systematic review and meta-analysis by Powchareon et al. demonstrated a reduced ischemic time to free flaps, reconstruction time, total operative time, and postoperative hospital stay [2]. The application of CAS also reduces the burden of decision-making intraoperatively, which helps to achieve adequate tumor resection margins [3]. By performing virtual surgical planning, the accuracy of the reconstruction is increased [4] and inter-segmental bone gaps can be reduced to a minimum [5]. This could lead to a more predictable reconstructive contour and bone healing. A comprehensive approach was also developed to systematically assess the spatial deviation of the reconstructed mandible in a computer-assisted reconstruction [6]. With the increased efficiency and accuracy of jaw reconstruction, more complicated functional jaw reconstructions with dental and neurosensory rehabilitation have increased in popularity.

Another advantage of CAS lies in its value for medical education and surgical training. Compared to free hand surgery, junior and less experienced surgeons can achieve more consistently favorable tumor resections and reconstruction outcomes, which is important for teaching hospitals [3].

#### 2.1.3. Concerns about Computer-Assisted Surgery

While providing a high predictability and reproducibility, CAS is also challenged for its rigidity and difficult improvisation during the surgery as most of the planning is performed before the operation.

The first and the most significant concern is for the oncological safety of the predetermination of the surgical margins, especially in malignant jaw pathologies where positive and close margins can lead to compromised local-regional control of disease and patient survival [3]. In CAS, virtual resections are performed with reference to the clinical examination, types of pathology, and preoperative imaging, which might cause uncertainties during the surgery [7]. Positive intraoperative frozen sections are possible and preoperative resection and reconstruction plans will have to be adjusted accordingly [8]. A study on ameloblastoma patients also yielded a comparable margin status with or without CAS. Furthermore, the possible additional advantage of decreasing the occurrence of close or positive margins was reported with CAS when applied to the resection of benign intraosseous lesions [9]. Compared to benign jaw lesions, CAS for malignant tumors poses more challenges in terms of the fast progress of disease during the period of virtual surgical planning and soft tissue margin determination. As reported by Pu et al., compared to the determination of the bone margin purely from a CT scan, the integration of careful clinical examination, intraoral optical scanning, and MRI was warranted for a reliable soft tissue margin during virtual surgical planning [3]. The timing of the surgery also plays a crucial role whereby surgeon dominated planning and in-house printing are very helpful. With the above measures, the study showed no significant difference in the margin status and survival performance of patients suffering from oral cancer with or without CAS after the adjustment of other significant risk factors [3]. 

Another concern is the low adherence to the CAS planning when the unexpected change of surgical plans arises intraoperatively. Wilde et al. [10] and Ma et al. [11] reported the need for an intraoperative change of plans in 19% and 17.6% of cases, respectively. However, a significantly lower rate of 5.1% was reported by Pu et al. [1] Different teams have different planning protocols which can lead to the significantly different performance of CAS [12]. Preoperative patient assessment and case selection play important parts as margin determination might be more difficult from imaging in certain cases, such as osteonecrosis of the jaw and tumors with a perineural invasion tendency. For oncological patients, timely surgery and confirmation of the surgical plan by the responsible surgeon before proceeding to surgery are crucial. A postoperative evaluation by comparing the preoperative plan to the final surgical outcomes with the feedback from the surgical team facilitates the rapid improvement in CAS skills.

The steep learning curve of CAS has also been regarded as a challenge when popularizing the technique. A cumulative sum analysis revealed a three-stage learning curve of CAS, including initial learning, plateau, and overlearning, and surgical proficiency was obtained after 23 cases [13]. This can guide the teaching and training of CAS.

Currently, CAS is used mostly for bony reconstruction. The computer-assisted reconstruction of soft tissue defects after the resection of malignant tumors is still under investigation and will be discussed in Section 3.3.

### 2.2. Patient-Specific Implants (PSI)

A significant improvement in bone fixation plates has been observed in the past decades. The traditional mass-produced fixation plates come with universal shapes. Bending and adaptation to the specific defect may be technique sensitive and time-consuming, especially for complex cases [14]. The repetitive bending may also decrease the fatigue and corrosion resistance of the plates leading to a higher risk of plate fracture, screw loosening, and bone resorption [15]. These complications led to the development of patient-specific implants which are prefabricated to fit the specific shape of the ideal reconstructed jaw.

Different types of patient specific implants have been fabricated so far, the most common being the patient-specific fixation plates. Other applications include various types of prostheses used for the reconstruction of the midface, mandible, and temporomandibular joint [16]. However, the application of these prostheses is currently limited to jaw defects after the ablation of benign lesions with a good soft tissue envelop. The problems of the fatigue and fracture of prostheses after long-term use and exposure though the skin after radiation therapy remain to be solved. In situations where implantable PSIs are not available, jaw models can be printed, and the surgeons can pre-bend the plates before the operation [17].

There are two main technologies for the fabrication of PSI: subtractive manufacturing (CAD/CAM) and additive manufacturing (3D printing) (Figure 1).

#### 2.2.1. CAD-CAM Patient-Specific Implants

The development of PSIs started with the popularization of the technology of computer-assisted design and computer-assisted manufacturing (CAD-CAM).

The traditional CAD-CAM technique was developed in the 1960s and started to be used in medical care in the 2000s, which led to a paradigm shift in head and neck reconstruction [18]. It led the frontier of osseous jaw reconstruction in the 2010s with its precision and intraoperative efficiency [19,20].

However, traditional CAD-CAM fabrication by subtractive milling from a block of material by computer numerical control (CNC) causes material waste. Certain complicated shapes cannot possibly be manufactured by subtractive milling, which led to the introduction and development of additive manufacturing, i.e., 3D printing [21].

#### 2.2.2. 3D-Printed Patient-Specific Implants

The three-dimensional (3D) printing technique, also known as additive manufacturing, experienced its significant development phase in the 2010s. Compared to CAD-CAM, 3D printing is a form of additive manufacturing that produces solid objects by adding materials layer by layer from base to top [22]. It offers a more versatile solution for complex structures and causes less material loss. However, some 3D printing technologies can be time-consuming, and the machines have a relatively high initial cost. Unlike objects made from a material block by CAD-CAM, 3D-printed objects cannot accommodate a high internal stress during manufacturing, which can lead to cracking and easy fracturing under functional stress [21].

Different types of 3D printing technologies have been developed. Each has its own advantages and disadvantages. The commonly used ones in the medical field include stereolithography (SLA) with liquid resin and selective laser melting (SLM) with powder materials [22]. In jaw reconstruction, the commonly used PSIs include reconstruction plates with Titanium, contour augmentation with porous polyethylene (e.g., Medpor), or polyetheretherketone (PEEK) [23]. 

#### 2.2.3. Advantages and Disadvantages of PSI

Compared to commercial stock plates, surgeons can avoid the time-consuming procedure of bending plates intraoperatively and avoid the risk of fatigue-induced plate fracture from repeated reverse bending. In combination with patient-specific osteotomy guides, PSIs can save a significant amount of intraoperative time which was previously used to segmentalize and adjust the bone segments of a bone graft (e.g., fibula flap) to produce an appropriate jaw contour. Yang et al. and Rana et al. reported increased accuracy with the use of PSIs in computer-assisted surgery [23,24]. With the increased efficiency and accuracy, PSIs bring opportunities for functional reconstruction such as immediate dental rehabilitation by simultaneous insertion of dental implants.

However, at the moment, an implantable PSI is still relatively expensive, and the printing technology may not be available in some parts of the world. Careful post-printing treatments are needed to reduce the chance of infection due to the inborn rough surface and plate fracture under stress due to possible microcracks inside the PSI [25]. 

#### 2.2.4. Guidelines and Regulations

With the popularization of 3D-printing technology, more 3D-printed medical devices are being adopted at the point of care (PoC). The recent systematic review by Murtezani et al. showed that 35% of studies were based on POC production methods while 12% were outsourced [26]. This has provided the timely production of devices suitable for specific clinical use. However, this also causes new challenges for the regulatory bodies’ s ability to ensure the safety and effectiveness of 3D-printed medical devices. A discussion paper was published in December 2021 by the U.S. Food & Drug Administration to seek advice from health care providers, facilities, medical device manufacturers, and other stakeholders in order to form guidelines and regulations for future use [27]. 

### 2.3. Dental Rehabilitation

Traditionally, dental rehabilitation was performed as a secondary procedure after the primary reconstruction of the jaw was completed, at least 3 to 6 months after the primary surgery. Besides the long waiting time until the patient can have a functional occlusion, this technique has several other disadvantages. If dental rehabilitation was not taken into consideration when designing the osseous flaps, the bony segments could be placed at an unfavorable position and angle, making further dental rehabilitation difficult, if not impossible. Moreover, when patients have undergone radiation therapy after the resection and reconstruction of the jaw, the placement of osseointegrated dental implants always carries the risk of osteonecrosis of the jaw and a loss of bone flap in the long term.

With the development of CAS and PSI, the accuracy and efficiency of jaw reconstruction have been significantly increased [28]. On top of restoring facial aesthetics and maintaining the airway, the timely and predictable restoration of the mastication function has become the new aim of functional jaw reconstruction. Different techniques of dental rehabilitation with osseointegrated dental implants in the reconstructed jaw have been reported.

Schepers et al. described a technique comprising the secondary reconstruction of the jaw using prefabricated fibula grafts with pre-placed dental implants [29]. In the first stage of surgery, dental implants were placed into the fibula with prefabricated guides and left in situ for osseointegration. A CT scan of the fibula was performed, and the reconstruction of the jaw was planned virtually with reference to the implants placed in the fibula. A second surgery was performed for the reconstruction of the jaw with the delivery of a dental prosthesis. This technique has the advantage of reducing the effect of errors in the placement of dental implants in the fibula. However, it requires multiple operations and is thus not popularized in most centers.

Levine et al. proposed the concept of “Jaw-in-a-day”, where dental implants and dental prostheses were placed at the same stage of the primary reconstruction of the jaw [30]. As the preliminary report, it proved the effectiveness of performing immediate dental rehabilitation at the same stage of tumor resection and reconstruction. The detailed total virtual workflow was further described by Zweifel et al. in 2018 [31].

When the immediate delivery of a dental prosthesis is planned, a higher accuracy for the jaw reconstruction and implant placement is required. Multiple attempts have been made to improve the accuracy of simultaneous dental implants. A tooth-borne or plate-borne implant position verification guide was developed by Zweifel et al. [32] However, this technique cannot be applied when the patient’s preexisting or remaining teeth are less than ideal. Schepers et al. used an occlusal splint to locate the implant-borne prosthesis when fixing the reconstruction segments [33]. To use this technique, an accurate jaw relationship registration is important, but it is often difficult, especially in oncological patients where preoperative occlusion is deranged due to the pathology. A “three-in-one” patient-specific surgical guide was reported by Zhu et al. to serve the purpose of fibula harvesting, segmentation, and simultaneous dental implant placement [34]. To overcome the sliding and rotating errors caused by the placement of fibula cutting guides, Pu et al. developed a novel malleolus cap for fibula flap harvesting. With the use of a malleolus cap, the simultaneous dental implants in the fibula approached a similar level of accuracy compared to the guided implant placement in the native maxilla and mandible, which further proved the reliability of the jaw-in-a-day technique with simultaneous dental rehabilitation [35]. 

### 2.4. Neurosensory Reconstruction

#### 2.4.1. Mental Nerve Reinnervation

In most cases of mandibulectomy, the inferior alveolar nerve (IAN) is sacrificed due to oncological safety concerns or technical difficulties. Without proper reconstruction, the spontaneous recovery of the IAN has been shown to be minimal and rarely functionally meaningful [36]. The loss of IAN can lead to anesthesia of the lower lip and chin resulting in functional impairments such as drooling, lip biting, and accidental damage to the skin of the lower chin during shaving in men, as well as uncomfortable feelings of dysesthesia and hyperesthesia which affect eating, talking, and quality of life [37]. The repair or reconstruction of the IAN after mandibulectomy has been attempted since the 1970s [38]. However, the reconstruction of the IAN was time-consuming, especially in cases with lengthy resection and reconstruction. With the improved efficiency and accuracy of jaw reconstruction through the application of CAS, neurosensory rehabilitation has regained attention in recent years. Attempts have been made to find better methods of nerve reconstruction with more predictable results and less donor site morbidity.

Autograft has been considered the “gold standard” for its well-proven clinical performance [39]. The greater auricular nerve, sural nerve, forearm cutaneous nerves, and long thoracic nerves were once used as the donor nerve for the nerve graft, but each has its own morbidity [40,41,42,43,44]. Nerve conduits have been used in IAN repair, but the clinical results are not consistent in the literature [45]. In recent years, processed nerve graft (PNA) has been reported to have promising results. In the Registry of ADVANCE Nerve Graft Evaluation Utilization and Outcomes for Reconstruction of Peripheral Nerve Discontinuities study, 87% of clinically meaningful recovery was observed in a total of 76 patients [46,47]. Miloro et al. also applied the technique of VSP preoperatively to predict the length of PNA required [48]. However, PNA products are not readily available in some countries.

Besides the selection of the nerve grafting material, there have been advances in techniques for autograft repair of the IAN after mandibulectomy. In the earlier studies, autografts have been anastomosed between the proximal and the distal ends of the IAN at the recipient site. However, for extensive oncological resections, the proximal end of the nerve may need to be sacrificed. The large gap also poses an additional risk for nerve grafts in terms of the recovery of sensation, as the clinical results of long-span nerve grafts are still under validation [49]. Moreover, after the inset of the fibula free flap, the nerve graft usually sits on the superior surface of the fibula segments for the best physiological positioning. This poses additional risks of accidental damage to the nerve during second-stage surgeries such as dental implant placement or vestibuloplasty. To solve these problems, the cross-face nerve graft has emerged as a new technique. In practice, the distal end of the damaged nerve was grafted and connected to the contralateral donor nerve crossing the face [50]. Catapano et al. applied this technique to reinnervate the mental nerve using the sural nerve as the donor nerve [51]. The results are preliminary with a small sample size and short follow-up. However, this opened new opportunities as the nerve coaptation to the contralateral mental nerve is easier to perform due to its superficial location and its being unaffected by the resection margin at the proximal end of the recipient’s nerve. Microsurgical techniques of end-to-side anastomosis are required for the cross-face nerve graft to avoid damage to the contralateral mental nerve.

#### 2.4.2. Sensate Osteocutaneous Flap

Attempts have been made to restore the oral sensation at the same stage of jaw reconstruction. Boyd et al. investigated the neurosomal anatomy of the fibula free flap skin paddle and discovered the dual innervation of the lateral sural cutaneous nerve (LSCN) and recurrent superficial peroneal nerve (RSPN). The same group also explored the possibility of anastomosing these nerves into the recipient site for jaw reconstruction using a sensate osteocutaneous flap [52,53]. 

In the case report by Tanaka et al., both sensate skin paddle and the reinnervation of the bilateral mental nerve were achieved at the same stage by end-to-end anastomosing the sural nerve graft included in the free fibular flap skin paddle to bilateral proximal ends of the IAN and end-to-side anastomosing the remaining mental nerves to the sural nerve [54]. Although the technique needs to be verified with a larger sample size, this further pushed the boundaries of neurosensory jaw reconstruction.

## 3. Future Perspectives

### 3.1. Augmented Reality (AR) and Mixed Reality (MR)

The technology of augmented reality, also called mixed reality, has been used in jaw reconstruction surgery in recent years to overcome the shortcomings of conventional surgical navigation. When applying the conventional navigation system, the surgeons need to repeatedly switch attention from the surgical field to the 2-dimensional screen of the navigation system. This system provides only limited 3D information and relies entirely on the hand-eye coordination of the surgeon to correlate the real situation in the surgical field and the feedback from the navigation system on the screen. AR can overlap the 3D image pre-registered from the patients’ imaging (and surgical planning when appropriate) onto the intraoperative surgical field.

There are currently two main streams of intraoperative AR, one with a monitor presenting the patient’s data with the 3D model overlaid, and the other with a see-through head-mounted display where the real world and the 3D projected image are overlaid and viewed by the surgeon through the glasses, e.g., HoloLens (Microsoft Corporation; Redmond, WA, USA). The different levels of clinical acceptance for these techniques were investigated through surgeons’ perspectives [55]. 

Compared to conventional surgical navigation, AR provides a more intuitive guidance system with better depth perception and hand-eye coordination, which is especially helpful for inexperienced surgeons [56]. It facilitates the development of template-free surgeries in mandible angle osteotomy [57] and waferless maxilla repositioning [58]. With reliable AR technology, the development of semi-automatic robotic surgery is also around the corner [59]. 

While AR has been applied for various purposes, including resections of tumors positioned at jaws, dental implant placement and root canals, cranial vault surgery, etc., [60] the application of AR/MR in jaw reconstruction is still in the pre-clinical development stage. The cadaveric study by Meng et al. investigated the feasibility of using MR in mandible reconstruction with fibula flap. Although the mean deviation of the osteotomies was at 2 mm, the intergonial angle deviation was reported to be 7 mm. The study also revealed the difficulties in precise registration, accurate control, and time concerns [61]. Yang et al. used the MR technology in four clinical cases of jaw tumor resection and reconstruction and the error was reported to be less than 4.79 mm in most areas, which is less than ideal if simultaneous dental rehabilitation is planned [62]. Battaglia et al. preliminarily reported three cases using AR technology in free fibula bone harvesting with reference to the CAD-CAM fibula segmentation guides, but no accuracy data were provided [63]. The same group subsequently investigated the accuracy of a marker-less AR-based protocol for skin paddle harvesting in phantom models. Optimal lighting conditions and a further improvement in marker-less tracking technologies are essential for clinically applied AR-assisted reconstructive surgery in the future [64]. 

The technology of AR has also been used to verify the location of dental implants inserted into the reconstructed jaw by augmenting the planned implant location and final prosthesis to the surgical field using a smartphone and a specially designed program. The alignment was still relying on the registration of a coded block attached to the surgical guide mounted onto the existing dentition [65]. The main difficulty of this process lies in the accurate and timely registration and tracking of the model to the real anatomy intraoperatively. There is more work to be done before this technique is to be widely used in the operating theater with clinically acceptable accuracy and predictability.

### 3.2. Artificial Intelligence (AI)

Computer-assisted jaw reconstruction offers the advantages of high accuracy and intraoperative efficiency. However, the process of pre-operative computer planning is time-consuming and technique sensitive.

Recent years have witnessed the great development of artificial intelligence and machine learning. However, in the medical field, the majority of the research has been focused on disease diagnosis and prognosis [66,67]. In the field of jaw reconstruction, research on AI is scarce. Early attempts have been made on AI-assisted segmentation from imaging and model building. The preliminary study by Yang et al. verified the reliability of the AI-assisted program in Mimics Viewer (Materialise, Belgium) for the segmentation of different structures such as the orbit, jaw, teeth, etc., [68]. Model building is the first step of computer-assisted surgery. Based on this, further efforts can be made to apply AI technology to assisting the virtual tumor resection, bone graft folding and alignment, patient-specific plate design, and dental implant placement.

### 3.3. Virtual Surgical Planning (VSP) for Soft Tissue Reconstruction

Soft tissue reconstruction for maxillofacial defects plays a major role in facial aesthetics. With the dental implantation in the reconstructed jaw becoming a new norm, soft tissue condition around the dental implants is crucial for the management of peri-implantitis and maintaining the quality of life of patients in the long run.

Although current VSP techniques are relatively mature for the hard tissue reconstruction of the jaw, soft tissue reconstruction with the skin paddle is largely arbitrary. On the one hand, this is due to the unpredictable soft tissue defect after the tumor resection, which is more difficult to delineate in the preoperative imaging compared to the bony defect. On the other hand, soft tissue’s shape and size change significantly over time, especially in patients with postoperative adjuvant radiation therapy. Attempts have been made to design the fibula skin paddle preoperatively based on the perforators shown in the CT angiogram. A skin paddle-outlining guide was used to assist in harvesting the fibula skin paddle [69]. However, this technique is only suitable in cases where there is a limited soft-tissue defect and the long-term performance of the designed skin paddle around dental implants is largely unknown.

For a reliable soft tissue VSP, more data is required on the aspects of a predictable virtual surgical planning for the soft-tissue defect, postoperative volumetric change of reconstructed soft tissue, and long-term performance of the skin paddle around dental implants in the reconstructed jaw.

### 3.4. Tissue Engineering and Bioprinting in Jaw Reconstruction

While the autogenous bony tissue remains the mainstream of jaw reconstruction after ablative surgeries, donor site morbidity, prolonged operation times, and hospital stay cannot be ignored. Moreover, autogenous tissue is limited by its shape and may not exactly mimic the existing jaw.

Efforts have been made in recent years to apply tissue engineering to jaw reconstruction. In the case series reported by Melville et al., mandible defects were successfully reconstructed by bone tissue engineering with a mixture of bone marrow aspirate concentrate (BMAC), bone morphogenic protein (BMP), and particulate allogeneic bone grafts, contained in a Ti mesh or resorbable membrane [70]. (Figure 2) With a similar principle of bone tissue engineering, Schlund et al. used fresh-frozen humeral allograft as the scaffold for the reconstruction of a posttraumatic mandibular defect [71]. 

The technology of bioprinting has also been under investigation in multiple animal studies to serve as a scaffold for bone engineering with or without bioactive molecules [72]. The most commonly used materials include biodegradable polylactic acid, polycaprolactone, calcium phosphate salts including hydroxyapatite, and beta-tricalcium phosphate (TCP) [73,74]. The combination of TCP with BMP was proven to be effective in bone tissue engineering in primates [75]. 

The success of a tissue-engineered bone graft relies significantly on the condition of the neighboring soft tissue for satisfactory blood supply and the strict separation from the oral cavity to avoid contamination. Currently, bone tissue engineering has mostly been applied to benign mandibular defects with no significant soft tissue defect from the ablative surgery [70]. Although promising, the technique is purely dependent on the soft tissue of the host as a “bioreactor” [76]. The ability to regenerate bone gives the surgeon another avenue to reconstruct the patient to normalcy without donor site morbidity [70]. To address instances where the soft tissue condition is suboptimal or a soft tissue defect is present after the excision of a malignant tumor, a combination of tissue-engineered bone with radial forearm flap and latissimus dorsi flap has been reported by Schlund et al. [71] and Ismail et al. [77], respectively. (Figure 3) Soft tissue engineering was also applied to fabricate keratinized oral mucosal grafts which were pre-laminated to a fibula flap for later jaw reconstruction [78]. This keratinized mucosa might show advantages over traditional skin paddles when dental implants are inserted in the future. As biotechnology evolves, the authors anticipate an advancement to prefabricated customized osteocutaneous flaps customized to the patients’ defects [79].

Regardless of the exciting potential of tissue engineering for replacing the traditional autogenous bone grafts and free flaps, several questions remain unanswered. Significant resorption was reported by Ismail in 2021 when the tissue-engineered bone block was transferred to the latissimus dorsi flap site. The rate of degradation and long-term stability remains unknown. With the relatively short history of clinical application, the possible immunologic reactions are still yet to be reported. Although the combination of BMA, BMP, and allogenic bone grafts have proven to be effective in bone tissue engineering, the optimal proportion and form of the components still remain to be investigated before this technique can be popularized in the future.

### 3.5. Muscular and Temporomandibular Joint (TMJ) Function

With the ever-rising standard of functional reconstruction, masticatory rehabilitation has become one of its main aims. However, the imbalance of masticatory muscles after jaw resection and reconstruction could lead to a TMJ dislocation, a low chewing efficiency, and a compromised quality of life [80]. The accurate location of the reconstructed bone and dental implants is only the first step. The location and movement of temporomandibular joints under the influence of masticatory muscles play an important role in the function of the reconstructed stomatognathic complex.

Bai et al. investigated 30 patients who underwent mandibulectomies involving the condyle for benign tumors and reconstruction with free fibular flaps. A significant decrease in the total volume of the masticatory muscles was observed on the affected side. Different muscles showed different reattachment patterns. Most of the cases achieved a lateral pterygoid muscle reattachment within 6 months, and an ectopic attachment of medial pterygoid muscles occurred in all cases. However, masseter reattachment on the affected side was only achieved in 10% of cases [81]. No data were available on how these affected the biting force, chewing efficiency, and quality of life of the patients.

The deviation in the static location of the temporomandibular joints after unilateral jaw reconstruction was investigated by Yang et al. Patients with condyles removed had higher deviations in the condyle and joint space, and 3D printed patient-specific plates increased the spatial accuracy of TMJ reconstruction [82]. More research is required to reveal how the reconstructed TMJ adjusts to the newly reconstructed jaw in the long term.

Based on more sound knowledge of masticatory muscle and TMJ function after jaw reconstruction, a more comprehensive and evidence-based stomatognathic system training and rehabilitation can be proposed in the future.

## 4. Conclusions

The reconstruction and rehabilitation of jaw defects have been revolutionized in recent years by the development of computer-assisted surgery and virtual surgical planning. The surgeons can restore the function and aesthetics at the same stage of ablative surgery with the application of patient-specific implants and the jaw-in-a-day technique. Currently, a great deal of research is being conducted on augmented/mixed reality, artificial intelligence, virtual surgical planning for soft tissue reconstruction, and the rehabilitation of the stomatognathic system. This will lead to an even more exciting future for the functional reconstruction and rehabilitation of jaw defects following ablative surgery.

## Figures and Tables

**Figure 1 cancers-14-03308-f001:**
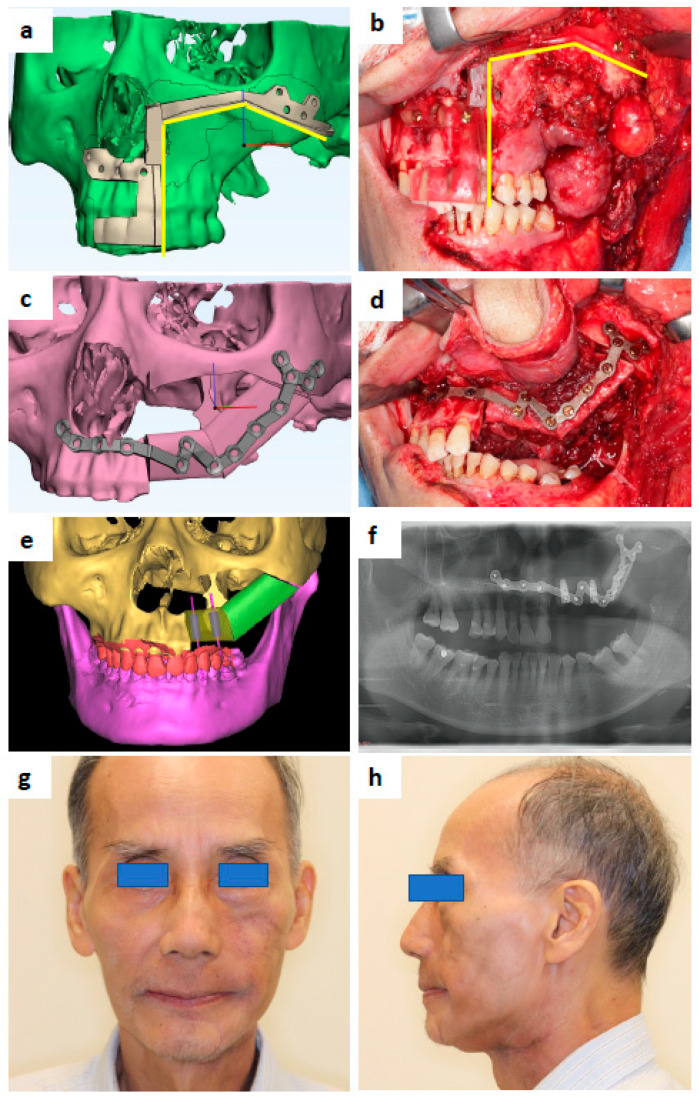
A 69-year-old male that presented with carcinoma ex pleomorphic adenoma at the left maxilla. (**a**) Maxilla resection guide design. (**b**) 3D-printed maxilla resection guide fitted intraoperatively. (**c**) Patient-specific Titanium plate design. (**d**) 3D-printed Ti plate fitted intraoperatively. (**e**) Design showing the location of simultaneous dental implants to be inserted during fibula free flap harvest. (**f**) Post-operative orthopantomography. (**g**) Postoperative 7 months and post-radiation 4 months—frontal view. (**h**) Profile view.

**Figure 2 cancers-14-03308-f002:**
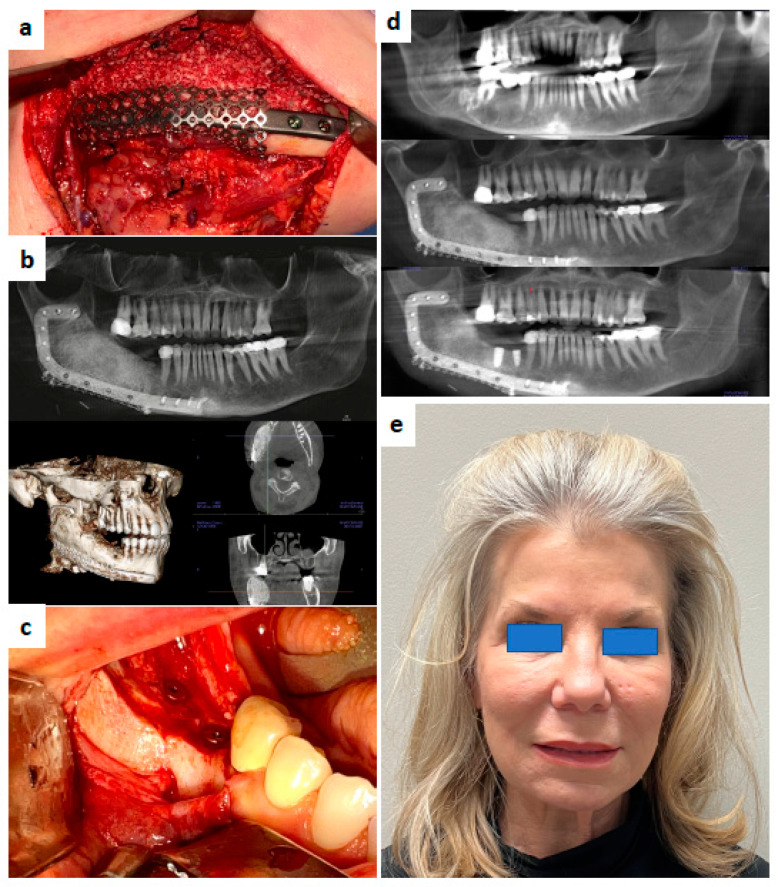
(**a**) A 61-year-old female presented with recurrent odontogenic keratocyst after 9 enucleation-curettage procedures. Resection was performed with immediate placement of Tissue Engineered Graft (BMAC + rh-BMP2 + Allogeneic Bone). (**b**) Eight-month CBCT demonstrating excellent bone regeneration and density. (**c**) Excellent regeneration of mandible with neo mental nerve foramen (Axogen Advance nerve graft placed at the same time of bone reconstruction). Regenerated bone was leveled off due to too much bone regeneration and appropriate occlusal space was restored before placement of dental implants. (**d**) Serial panorex (top to bottom) original lesion on right of mandible, with a 6 month follow up and 1.2 months after placement of dental implants. (**e**) Patient with normal facial contours and function 1 year after the surgery.

**Figure 3 cancers-14-03308-f003:**
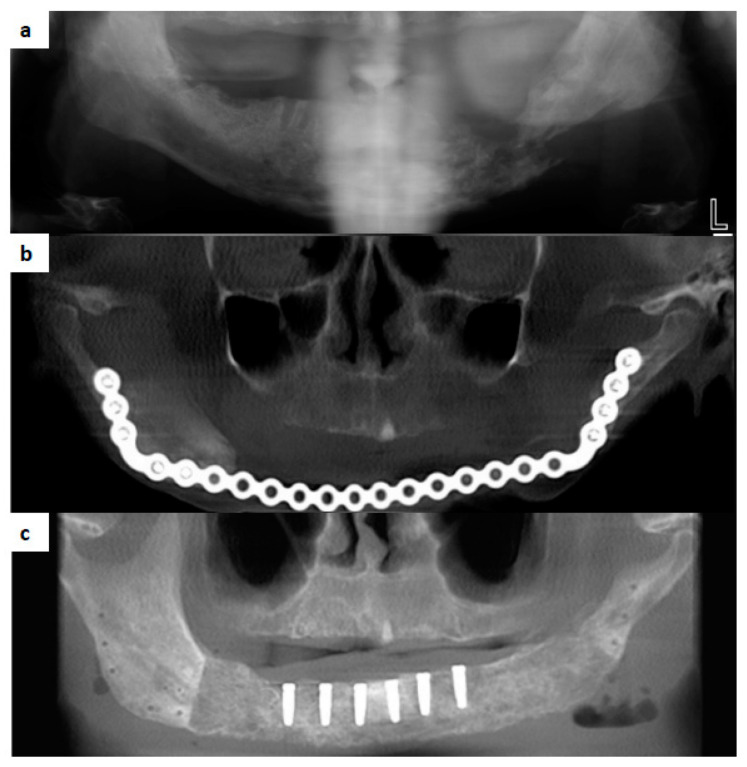
(**a**) A 64-year-old male with >7000 cGy radiation for oral cancer, who subsequently developed osteoradionecrosis of the jaw. Patient had bilateral arthrosclerosis of the peroneal artery. (**b**) Patient reconstructed with ALT and mandibular plate. (**c**) Patient reconstructed to full dental rehabilitation with a neo regenerated mandible (rh-BMP2. allogeneic bone, and bone marrow aspirate BMAC).

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
