# Peer review of "Current Trends in the Reconstruction and Rehabilitation of Jaw following Ablative Surgery"

_cancers, 2022, doi:10.3390/cancers14143308_

Round 1

Reviewer 1 Report

Well organized and comprehensive review regarding to the jaw reconstruction after tumor ablative surgery. No further comment on the present form.

Author Response

Response to Reviewer 1 Comments

Well organized and comprehensive review regarding to the jaw reconstruction after tumor ablative surgery. No further comment on the present form.

Response: Thank you very much for reviewing our article.

Reviewer 2 Report

Well written. Covers contemporary new developments in jaw recon. I’d consider removing the nerve paragraph since this may not be s new.

Author Response

Response to Reviewer 2 Comments

Well written. Covers contemporary new developments in jaw recon. I’d consider removing the nerve paragraph since this may not be s new.

Response: Thank you very much for the suggestion. We have made the nerve reconstruction paragraph more concise and focused more on the new knowledge of this aspect.

Reviewer 3 Report

The article provides a broad summary of the recent technological innovations and the prospects concerning jaws reconstruction after demolitive surgery. The maxilla and the mandible play a crucial role in breathing, chewing, swallowing and speech. Ablative surgery does not only impact on aesthetic outcomes, but it might affect all these vital functions. The article fluently encompasses most of the recent high-tech instruments in use, like CAD-CAM, VSP and PSI, to reconstruct the jaws and how their introduction has impacted on the surgical final outcomes, in terms of function, timing, patient satisfaction and surgeons learning curve. The authors also provide a comprehensive review of the future perspectives, such as augmented reality, mixed reality, tissue engineering and bioprinting and how these techniques could improve surgery in the future. Even though, more and more articles on these topics are becoming popular in literature, this paper gives the reader the possibility to have a global vision of the techniques available, their main advantages and disadvantages.

The manuscript is clear and easy to read. The figures show clinical cases and provide an intuitive overview of the methods employed. The conclusions are consistent with the evidences and arguments presented addressing the main topic posed.

However, according to this review, some fundamental aspects should be addressed. They will be listed below:

Major issues:

·         Research methods. How the relevant literature was identified should be specified in terms of keywords and relevant sources used and according to what they were evaluated and selected. If the authors intent was to present a Systemic Review the PRISMA protocol must be included. If the aim was a literature review, a brief paragraph including methods of research would be enough. The lack of these information inevitably impacts on the article reliability.

·         Tumor entity. The authors should try to distinguish the use, pros and cons of the different techniques considering the type of pathology and its nature. Especially, with regard to tumors, the difference between malignant and benign pathologies should be underlined. The possible limitations of these techniques when dealing with malignant tumors should not be discarded.

·         Techniques description. The authors should consider that not all the readers have knowledge of these new techniques. A brief description of the different methods should be included, to make the article more comprehensible and easier to read.

·         Abstract. The abstract should include the aim and the type of research, in order to let the readers immediately get the essence of the paper.

Minor issues:

·         2.4 Neurosensory reconstruction. The article should clarify the role of these technological innovations in nerve reinnervation and what could be their advantages if compared to traditional surgical techniques.

Other points:

Custom-made titanium mandible prosthesis. A brief mention to the mandibular reconstruction using custom-made titanium prosthesis would be appreciated in the setting of a such comprehensive article. Even though, their use is limited, they still represent an available option in mandibular reconstruction

Author Response

Response to Reviewer 3 Comments

Research methods. How the relevant literature was identified should be specified in terms of keywords and relevant sources used and according to what they were evaluated and selected. If the authors intent was to present a Systemic Review the PRISMA protocol must be included. If the aim was a literature review, a brief paragraph including methods of research would be enough. The lack of these information inevitably impacts on the article reliability.

Response: Thank you very much for the suggestion. We have added a paragraph describing the method of search.

Tumor entity. The authors should try to distinguish the use, pros and cons of the different techniques considering the type of pathology and its nature. Especially, with regard to tumors, the difference between malignant and benign pathologies should be underlined. The possible limitations of these techniques when dealing with malignant tumors should not be discarded.

Response: Thanks for the comment. We have revised 2.1.3 discussing the use of CAS in malignancy cases especially regarding the oncological safety and the soft tissue defect; and 3.4 discussing the application of tissue engineering in jaw defects after resection of malignant jaw tumors.

Techniques description. The authors should consider that not all the readers have knowledge of these new techniques. A brief description of the different methods should be included, to make the article more comprehensible and easier to read.

Response: Thank you very much for the suggestion. The description of certain techniques was added accordingly.

Abstract. The abstract should include the aim and the type of research, in order to let the readers immediately get the essence of the paper.

Response: Thanks for the suggestion. The aim and type of research were added to the abstract.

Neurosensory reconstruction. The article should clarify the role of these technological innovations in nerve reinnervation and what could be their advantages if compared to traditional surgical techniques.

Response: Thanks for the comment. We have revised 2.4 Neurosensory Reconstruction with the focus on recent innovations on the aspects of nerve graft material and technique.

Custom-made titanium mandible prosthesis. A brief mention to the mandibular reconstruction using custom-made titanium prosthesis would be appreciated in the setting of a such comprehensive article. Even though, their use is limited, they still represent an available option in mandibular reconstruction.

Response: Thank you very much for the suggestion. The information on patient specific prosthesis has been added to the section of 2.2 Patient-Specific Implants. The references have been adjusted accordingly.

Round 2

Reviewer 3 Report

After the first peer review, the authors made the required corrections. The article is now more appropriate and complete. No further adjustments are needed.